# Novel Insights into Psychosis and Antipsychotic Interventions: From Managing Symptoms to Improving Outcomes

**DOI:** 10.3390/ijms25115904

**Published:** 2024-05-28

**Authors:** Adonis Sfera, Hassan Imran, Dan O. Sfera, Jacob J. Anton, Zisis Kozlakidis, Sabine Hazan

**Affiliations:** 1Patton State Hospital, 3102 Highland Ave., Patton, CA 92369, USA; hassan.inran@dsh.ca.gov (H.I.);; 2University of California Riverside, Riverside 900 University Ave., Riverside, CA 92521, USA; 3Loma Linda University, 11139 Anderson St., Loma Linda, CA 92350, USA; 4California Baptist University, Riverside, CA 92521, USA; 5International Agency for Research on Cancer, 69372 Lyon, France; kozlakidisz@iarc.who.int; 6ProgenaBiome, Ventura, CA 93003, USA; drhazan@progenabiome.com

**Keywords:** aryl hydrocarbon receptor, dopamine, antipsychotic drugs, naturally occurring antipsychotics, senotherapeutics

## Abstract

For the past 70 years, the dopamine hypothesis has been the key working model in schizophrenia. This has contributed to the development of numerous inhibitors of dopaminergic signaling and antipsychotic drugs, which led to rapid symptom resolution but only marginal outcome improvement. Over the past decades, there has been limited research on the quantifiable pathological changes in schizophrenia, including premature cellular/neuronal senescence, brain volume loss, the attenuation of gamma oscillations in electroencephalograms, and the oxidation of lipids in the plasma and mitochondrial membranes. We surmise that the aberrant activation of the aryl hydrocarbon receptor by toxins derived from gut microbes or the environment drives premature cellular and neuronal senescence, a hallmark of schizophrenia. Early brain aging promotes secondary changes, including the impairment and loss of mitochondria, gray matter depletion, decreased gamma oscillations, and a compensatory metabolic shift to lactate and lactylation. The aim of this narrative review is twofold: (1) to summarize what is known about premature cellular/neuronal senescence in schizophrenia or schizophrenia-like disorders, and (2) to discuss novel strategies for improving long-term outcomes in severe mental illness with natural senotherapeutics, membrane lipid replacement, mitochondrial transplantation, microbial phenazines, novel antioxidant phenothiazines, inhibitors of glycogen synthase kinase-3 beta, and aryl hydrocarbon receptor antagonists.

## 1. Introduction

The discovery of chlorpromazine in the 1950s revolutionized psychiatry and contributed to the deinstitutionalization of people with severe mental illness. Subsequently, the homelessness and incarceration of individuals with schizophrenia (SCZ) and schizophrenia-like disorders (SLDs) dramatically increased, suggesting that symptomatic relief in these conditions rarely translates into sustained recovery [1,2].

Although most patients treated with antipsychotic drugs attain partial remission or amelioration of symptoms, few return to premorbid levels of functioning, measured by stable employment, attending school, raising a family, and being independent in all activities of daily living (ADLs) [3]. For this reason, large public institutions for the treatment of mental illness, such as state hospitals, are still in existence, while sanatoria for tuberculosis or leprosy were closed almost a century ago.

The early antipsychotic drugs were derived from methylene blue (MB), a phenothiazine synthesized in 1876 in Germany. The interest in this agent dramatically surged after the realization that it exerts antidepressant actions by inhibiting monoamine oxidase A (MAO-A), a surreptitious discovery that commenced the era of modern psychopharmacology [4]. Upon chlorpromazine’s approval in the US, over 40 dopamine-blocking antipsychotic drugs were developed, aiming at restoring premorbid functioning by targeting major symptoms.

Although antipsychotic drugs are extremely efficacious for acute psychosis as the symptoms are often cleared within hours or days, sustained recovery is achieved by only 13.5% of patients after an initial psychotic episode [5]. Moreover, looking at the entire 20th century, in the early decades, long-term recovery was at 20%, not differing from the end of the century when antipsychotic drugs were being widely utilized [6]. At present, 33% of patients with SCZ relapse within 12 months after an initial psychotic episode, 26% remain homeless at the 2-year follow-up, and 5 years after the first psychotic outbreak, only 10% are employed [7,8,9]. Together, these data indicate that the blockade of dopamine (DA) receptors seldom improves the outcomes of SCZ or SLDs. Moreover, some antipsychotic drugs, including clozapine and aripiprazole, upregulate DA, suggesting that this neurotransmitter may play an indirect role in the etiopathogenesis of severe mental illness [10,11].

The aims of this narrative review are as follows:To summarize what is known about the role of premature cellular/neuronal senescence in the pathogenesis of SCZ and SLDs.To discuss potential strategies for improving sustained recovery in SCZ and SLDs via natural senotherapeutics, microbial phenazines, aryl hydrocarbon receptor (AhR) antagonists, membrane lipid replacement (MLR), and mitochondrial transplantation.

## 2. Premature Cellular Senescence in Schizophrenia

Patients with SCZ and SLDs live, on average, 15–20 years less than the general population, exhibit shortened telomeres, and develop age-related diseases earlier in life, suggesting that premature cellular senescence plays an important role in this pathology [12,13,14,15]. Indeed, many researchers and clinicians refer to SCZ as a “segmental progeria” to highlight the accelerated aging of tissues and organs, including the brain, in this disorder [12].

Cellular senescence is a program of permanent cell cycle arrest with an active metabolism, shortened telomeres, accumulation of macromolecular aggregates, increased levels of senescence-associated β-galactosidase (SA-β-gal), and a toxic secretome, known as the senescence-associated secretory phenotype (SASP), which can spread senescence to neighboring healthy cells [16]. It is believed that cellular senescence defends against tumorigenesis by preventing oncogene-driven malignant transformation. However, the accumulation of aged cells and the subsequent inflammation may paradoxically promote cancer and disrupt biological barriers, facilitating the dissemination of metastases [16,17]. Inflammation and senescent cells increase the permeability of the gut barrier, facilitating the translocation of the gastrointestinal (GI) tract bacteria (or their molecules) into the systemic circulation, a phenomenon encountered in neuropsychiatric and neurodegenerative disorders [18]. In another example, microbiota-derived gallic acid converts p53, the key anticancer protein, into an oncogene that drives tumorigenesis [19,20,21]. As p53, an SCZ risk gene, also promotes cellular senescence, it likely connects microbial translocation to severe mental illness [22,23]. Indeed, bacterial molecules were demonstrated to induce cellular senescence in neurons and microglia, as documented in SCZ [24,25,26,27]. For example, *Escherichia coli* (*E. coli*)-induced psychosis was reported in epidemics as well as in urinary tract infections (UTI), connecting bacteria to SCZ and SLDs [28].

Recent studies have found that downregulated DA receptors and transporters induce premature neuronal senescence, suggesting that dopaminergic signaling is required to avert early brain aging [29]. These findings were further substantiated by virus-induced senescence (VIS), a phenotype documented during the COVID-19 pandemic, marked by brain aging due to the infection of DA neurons [29,30]. *Clostridium* sp. are gut microbes also known for interfering with dopaminergic signaling, further linking cellular senescence to insufficient DA [31,32]. This is significant since both treated and untreated SCZ patients were found to exhibit body-wide premature cellular/neuronal aging, linking this condition to abnormal intestinal permeability [33,34]. Indeed, SCZ has been associated with increased microbial migration into the host’s systemic circulation [35,36,37]. Furthermore, premature cellular senescence may contribute to the other SCZ markers, including gray matter loss, decreased mitochondrial abundance, the attenuation of gamma (γ)-oscillations in EEGs, and the peroxidation of cell membrane lipids.

### 2.1. Ferrosenescence vs. Ferroptosis

Fe^2+^ is an essential nutrient for both hosts and pathogens. It is also a cofactor in the biosynthesis of tyrosine hydroxylase and tryptophan hydroxylase, enzymes involved in the synthesis of DA and serotonin (5-HT), respectively. In the gut, DA acts as a microbial siderophore, which clears Fe^2+^ from the microenvironment, lowering the risk of ferroptosis [38,39].

Senescent cells, including neurons, upregulate intracellular Fe^2+^, which in the vicinity of cytosolic lipids, may increase the risk of peroxidation and neuronal death by ferroptosis. In addition, senescence-upregulated lactate may also increase intracellular Fe^2+^ levels further, predisposing to neuronal demise [40]. Moreover, Fe^2+^ and lactate are known to enhance the biosynthesis of ceramide, a cell membrane lipid, which, in excess, can cause neurotoxicity [41,42]. Ferroptosis has been documented in SCZ; however, senescent cells are often resistant to programmed cell death, suggesting that impaired autophagy may drive iron-mediated brain aging [43,44,45]. Conversely, inducing ferroptosis in senescent cells precipitates their clearance by the immune system, indicating that this mechanism may be compensatory [46].

In our previous work, we introduced the concept of ferrosenescence, senescent cells, including neurons, with damaged DNA and dysfunctional p53-mediated genomic repair, as well as defective NKCs that are incapable of clearing senescent cells [47,48,49]. Ferrosenescent cells are resistant to ferroptosis due to impaired ferritin autophagy as well as the upregulation of ferroptosis inhibitors [glutathione peroxidase 4 (GPX-4) and apolipoprotein E (APOE)] [50,51,52,53]. For example, in SCZ, upregulated APOE, low ferritin levels, and increased intracellular Fe^2+^ levels likely reflect ferrosenescence [43,54]. Consequently, we believe that ferrosenescence may be more prevalent in SCZ than ferroptosis. Moreover, ferrosenescence may account for the other SCZ markers, including a decreased brain volume, γ-oscillations in EEGs, and mitochondrial dysfunction [55,56,57,58].

### 2.2. Senescent Gut Barrier

The relationship between the gut microbiome and senescent intestinal epithelial cells (IECs) is an emerging field that likely plays a major role in SCZ and SLDs [59,60]. For example, antibodies against translocated microbes, such as *Hafnei alvei*, *Pseudomonas aeruginosa*, *Pseudomonas putida*, and *Klebsiella pneumoniae*, were demonstrated in SCZ with negative symptoms, connecting microbial translocation to this pathology [61]. Moreover, translocated gut microbes may trigger nutritional immunity and iron sequestration in macrophages to withhold it from microbes, decreasing total circulatory Fe^2+^ levels [62]. In conditions of low circulatory Fe^2+^ levels, translocated bacteria may adopt a dormant state in human tissues, including the brain, awaiting increased Fe^2+^ availability to be reactivated [63,64].

The gut microbiota is immunologically tolerated in the gut lumen; however, upon translocation into the systemic circulation, the immune system is activated, triggering inflammation and, often, antibodies against microbial molecules. Since the microorganisms populating the GI tract express receptors identical or structurally related to human proteins, antibodies against these molecules may be construed as autoantibodies. Moreover, translocated microbes elicit inflammation, promoting cellular senescence, engendering a vicious circle in which senescent IECs facilitate the translocation of bacteria across the lamina propria, while inflammation triggered by these pathogens promotes further senescence [65,66,67].

Senescent cells were recently found to play a major role in the pathogenesis of inflammatory bowel disease (IBD), a condition marked by the translocation of microbes from the GI tract into the host tissues, including the brain [68]. This is further enhanced by the increased prevalence of SCZ in patients with IBD, emphasizing the role of the GI tract in severe mental illness [68,69]. Microbial translocation has been extensively studied in human immunodeficiency virus (HIV) infection, a condition marked by the massive exit of microbes from the GI tract due to interleukin-22 (IL-22) deficiency [70,71]. For this reason, we believe that recombinant IL-22 may comprise a new SCZ treatment [72].

## 3. Aryl Hydrocarbon, the Master Regulator of Cellular Senescence

The AhR is a ligand-activated transcription factor originally characterized as the receptor for dioxin (2,3,7,8-tetrachlorodibenzo-p-dioxin). Later, it was revealed that the AhR is also activated by various endogenous and exogenous ligands, driving multiple physiological processes and pathologies, likely including SCZ [73,74]. Aside from being the master regulator of cellular senescence, the AhR negatively regulates lactate and, by extension, the posttranslational modification, lactylation. As such, a dysfunctional AhR may drive both premature neuronal and glial aging as well as the excessive lactylation documented in SCZ and SLDs [75,76].

Several AhR ligands are molecules of interest in neuropsychiatry, including DA, phenazines, phenothiazines, serotonin, melatonin, and clozapine, suggesting that this transcription factor may play a significant role in the etiopathogenesis of SCZ (Figure 1) [77,78,79]. In prokaryotes, DA exerts iron-chelating properties; therefore, depletion in this neurotransmitter may upregulate intracellular Fe^2+^, leading to ferroptosis or ferrosenescence [39,43]. In addition, DA enhances the phagocytic properties of NKCs, facilitating the elimination of senescent and damaged cells [80]. Moreover, DA, via the DA 1 receptor (D1R), enhances the generation of acetylcholine (ACh), a neurotransmitter required for maintaining both the gray matter volume and rapid brain oscillations [81]. Conversely, lowering DA levels may promote inflammation by the accumulation of aged cells.

Compared with young neurons, senescent neuronal cells downregulate most surface receptors, including the dopaminergic and cholinergic ones, contain fewer mitochondria, and undergo metabolic and epigenetic reprogramming via lactate and lactylation, respectively [82]. Indeed, the lactylation of histone 3 (H3) lysine opposes neuronal senescence and restores the pre-senescent cellular status, suggesting that harnessing lactylation may comprise a potential neuropsychiatric therapy [83]. However, dysfunctional lactylation was shown to induce cell cycle reentry in senescent neurons, as demonstrated in Alzheimer’s disease (AD) and other neurodegenerative disorders [84]. Moreover, senescent microglia with histone 3 (H3K18) lactylation were demonstrated to adopt a neurotoxic phenotype, engaging in the elimination of healthy synapses and neurons, a phenomenon documented in SCZ and neurodegeneration [85,86,87,88,89]. Following this line of research, preclinical studies have revealed that neuronal excitation and social stress enhance the lactylation of histones, further linking this posttranslational modification to the pathogenesis of SCZ and SLDs [90]. Furthermore, previous studies have associated SCZ with increased brain lactate levels and the alteration of histone proteins, suggesting that lactylated neurotoxic microglia may be more detrimental to severe mental illness than previously thought [91,92,93,94].
Figure 1In the cytosol, the AhR is stabilized by two HSP90 molecules. DA, oxidized lipids (and toxic ceramide), clozapine, serotonin, melatonin, and vitamin D3 are AhR ligands [95,96]. Pollutants, such as phthalate and bisphenol A (BPA), are also AhR ligands. In contrast, aripiprazole binds to the AhR chaperone, HSP90. HSP90 prevents AhR’s entry into the nucleus where it drives the transcription of genes, including those for cellular senescence.
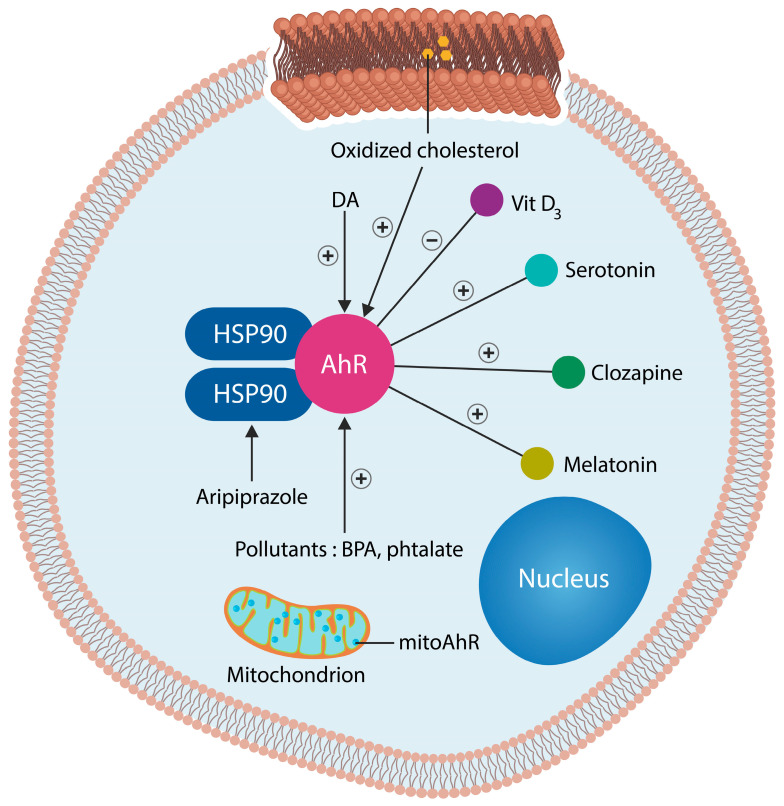



### 3.1. Gray and White Matter Loss

The data on sustained remission in SCZ match with neuroimaging studies, showing life-long gray matter loss in both medicated and unmedicated patients. This is significant, as preventing or restoring brain volume homeostasis likely improves outcomes in severe mental illness [97,98,99,100,101]. Indeed, preclinical studies have shown that AhR homeostasis is directly correlated with brain volume, suggesting that DA/AhR binding is critical for averting parenchymal loss [95]. The AhR also preserves the gray matter volume by regulating the availability of ACh, which, like DA, acts as a brain volume guardian [102]. Low brain DA levels also induce white matter loss, but to a lesser extent in SCZ compared with demyelinating disorders [103,104]. This is significant, as microbial toxins, including lipopolysaccharide (LPS), as well as environmental pollutants, such as plasticizers, are AhR ligands associated with SCZ. For example, the high comorbidity between SCZ and IBD may be the result of bacterial molecules “escaping” through the hyperpermeable intestinal barrier, a hallmark of IBD [105,106,107]. In addition, the higher prevalence of SCZ in northern regions of the world compared with the equatorial countries may be explained by AhR binding to vitamin D3 [96].

Taken together, DA is an indispensable neurotransmitter that prevents gray matter depletion by acting as an AhR ligand. Conversely, DA-blocking antipsychotic drugs may induce iatrogenic gray matter loss. This is further substantiated by the fact that clozapine, the most effective antipsychotic drug, is an AhR ligand that also upregulates DA levels [77,78,108].

### 3.2. Dopamine-Sparing Antipsychotics

Like SCZ itself, treatment with first- and second-generation antipsychotic drugs is associated with gray matter loss, suggesting that brain DA levels are strictly regulated so that minimal fluctuations in this neurotransmitter may cause brain volume depletion [99,100,109,110,111,112,113,114].

Novel studies have found that antipsychotic drugs transfer positive or negative electrical charges to their substrates, donating or accepting electrons. For example, DA, lithium, clozapine, novel phenothiazines, and aripiprazole give away electrons, preserving the gray matter volume, while most other antipsychotics are electron acceptors and associated with brain volume loss [115,116,117]. More studies are needed to identify and develop DA-sparing, electron-donor antipsychotics.

## 4. Mitochondrial Dysfunction and Loss of Gamma-Band Oscillations

According to the endosymbiotic hypothesis, mitochondria are derived from ancestral bacteria and can communicate with the gut microbes via chemical messengers, such as ROS and sphingolipids [118,119,120]. Ceramide, one of the sphingolipids, is secreted by the microbiome and, in excess, can be toxic to mitochondria [121]. Senescent cells, including neurons, upregulate ceramide, probably accounting for the paucity of mitochondria in the aging brain [122,123,124]. Moreover, senescence-upregulated intracellular Fe^2+^ disrupts ceramide metabolism, further contributing to mitochondrial loss [125]. Indeed, ceramide-induced mitochondrial damage has been associated with atherosclerosis and SCZ, disorders associated with premature cellular senescence [126,127].

Acid sphingomyelinase (ASM), the enzyme catalyzing the conversion of sphingomyelin into ceramide, has been identified as an SCZ target, and ASM inhibitors, such as fluvoxamine and rosuvastatin, appear to ameliorate clinical outcomes in this disorder [128]. In addition, a natural alkaloid, berberine, decreases ceramide levels as well as inflammation, indicating a potentially beneficial effect on SCZ and SLDs [129,130].

Neurons have the capability of replacing defective mitochondria by importing them from microglia and astrocytes via tunneling nanotubules or extracellular vesicles (EVs) (Figure 2) [131,132]. Indeed, preventing neuronal loss by supplying mitochondria to neurons is one of the main functions of astrocytes.

Interestingly, antidepressant drugs in the class of serotonin reuptake inhibitors (SSRIs), including fluvoxamine, facilitate mitochondrial export to neuronal cells, emphasizing the neuroprotective role of these agents (Figure 3) [133]. Indeed, SSRIs were demonstrated to delay the conversion of mild cognitive impairment (MCI) into dementia, indicating that mitochondrial import contributes to neuronal rescue [134]. In addition, ferroptosis-blocking drugs and iron chelators, including halogenated phenazines, may delay or prevent neurodegenerative disorders, suggesting a novel therapeutic approach (see the section on phenazines and phenothiazines) [135,136].
Figure 2Glial cells, including astrocytes, supply neurons with healthy mitochondria via tunneling nanotubules, preventing apoptosis [137,138]. In addition, astrocytes prevent neuronal ferroptosis by transferring antioxidants, including GPX-4. Astrocytes uptake cystine via the cystine/glutamate antiporter (Xc^−^). Cysteine can also be obtained from methionine via glutathione. Fe^2+^ enters neurons through the transferrin receptor-1 (TRF-1), which is stored in ferritin and requires ferritinophagy to be released. Excess Fe^2+^ exits the neurons via ferroportin (FPT) channels.
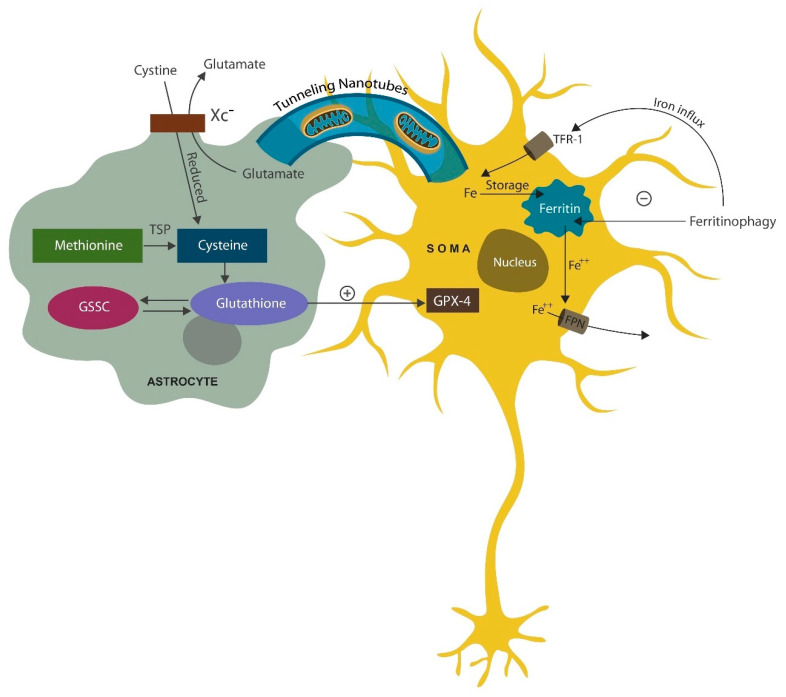



Aside from supplying healthy mitochondria to neurons, astrocytes also export antioxidants, including GPX-4, an enzyme involved in rescuing neuronal cells from ferroptosis. Additionally, ASM inhibitors, such as fluvoxamine, facilitate GPX-4 biosynthesis, averting ferroptosis. Indeed, fluvoxamine and other SSRIs preserve neuronal cells by both inhibiting ferroptosis and facilitating mitochondrial transfer [137,138]. GPX-4 is derived from cysteine, which enters astrocytes via the cystine/glutamate antiporter (Xc^−^) and enables the biosynthesis of glutathione and GPX-4 (Figure 2).

Given that γ-band frequencies are energy-consuming, mitochondria are essential for the generation of these rapid brain oscillations [139]. In this regard, preclinical studies have shown that lost γ-oscillations can be restored by inhibiting GSK-3β, an enzyme previously implicated in SCZ and SLDs [140,141]. Several antipsychotic drugs, including lithium, and natural compounds, kaempferol, are GSK-3β inhibitors and, therefore, capable of restoring the γ-rhythm (Figure 3).

### Entrainment of Gamma-Band Oscillations in Schizophrenia

Oscillations in the γ-range are rhythmic patterns of high-frequency (25 Hz to 100 Hz) EEG waves, playing a key role in cognition, attention, perception, and movement [142]. Under normal circumstances, this neuronal activity is synchronized across numerous brain regions, while in SCZ, there are γ-rhythm abnormalities, especially those elicited by auditory stimuli [143,144].

Interneurons, including parvalbumin (PV), vasoactive intestinal polypeptide (VIP), and somatostatin (SST), are the main drivers of γ-rhythms, indicating that the defective inhibition of pyramidal cells may trigger the loss of rapid rhythms [145,146]. Regarding neurotransmitters, γ-oscillations are dependent on ACh, an AhR-regulated biomolecule controlled by acetylcholinesterase (AChE), a direct AhR ligand [147].

Several studies have connected γ-oscillations to the microbiome-derived ACh, linking rapid EEG rhythms to intestinal microbes [148,149]. This is significant, as decreased brain ACh levels in SCZ have been associated with gray matter loss, emphasizing the key role of the AhR and AChE in maintaining both brain volume and rapid oscillations [150,151]. Given the important role of the cholinergic system in SCZ, it is not surprising that novel ACh-based antipsychotics have been developed [130,152].

Aside from ACh and mitochondrial import, lost γ-oscillations may be restored by entrainment with sensory stimuli, such as ultrasound or transcutaneous vagal nerve stimulation (tVNS) at 40 Hz, emphasizing a potential strategy for SCZ and SLDs [149,153]. Interestingly, tVNS improves not only neuronal function but also optimizes gut permeability, decreasing microbial translocation [149,154].

## 5. Senotherapeutics

It is currently established that severe mental illness is associated with cellular/neuronal senescence, indicating that endogenous or exogenous toxins may play a key role in this pathology [155,156,157]. For example, viral or bacterial infections induce premature aging in brain cells by eliciting immune responses, likely triggering psychosis [158,159]. Indeed, new-onset psychosis was documented in infections with senescence-inducing viruses including HIV and SARS-CoV-2 [160,161].

Senotherapeutics are natural or synthetic compounds that can delay, prevent, or reverse cellular/neuronal senescence. Senotherapeutics comprise senolytic agents that facilitate the elimination of senescent cells, and senomorphic compounds capable of deleting senescence markers, including SASP and SA-β-gal [162]. While, in the past, it was thought that cellular senescence could not be reversed, newer studies have found that inhibiting 3-phosphoinositide-dependent protein kinase-1 (PDK1) can revert cellular senescence in humans [163]. Interestingly, PDK1 is an upstream modulator of SCZ-linked GSK-3β.

Senolytic antibiotics belong to a distinct class of agents that include azithromycin, minocycline, and roxithromycin and possess neuroprotective, anti-inflammatory, and senolytic properties [164]. For example, it has been known for some time that minocycline may be beneficial for SCZ, suggesting that senolytics have a place in the treatment of SCZ and SLDs, probably by clearing neurotoxic glial cells [165].

Other senolytic agents relevant to SCZ and SLDs are summarized in Table 1.

A senolytic vaccine recently tested in progeroid mice may usher in a new era of neuropsychiatry, raising the possibility of vaccination or serum treatment for SCZ and SLDs [174]. Another immunological intervention, an antibody–drug conjugate against a membrane senescence marker, was demonstrated to clear senescent, damaged, or infected cells, emphasizing a new therapeutic strategy [175].

## 6. Membrane Lipid Replacement (MLR) 

MLR refers to the oral administration of natural cell membrane glycerophospholipids, along with kaempferol (3,4′,5,7-tetrahydroxyflavone), a flavonoid found in tea, broccoli, cabbage, kale, beans, endive, leek, tomato, strawberries, and grapes [176]. Like lithium and some antipsychotics, kaempferol is an inhibitor of GSK-3β, suggesting that it may exert antipsychotic properties without the typical adverse effects of psychotropic drugs [177,178].

The administration of MLR + kaempferol gradually replaces damaged phospholipids, ceramides, and oxysterols from neuronal and/or mitochondrial membranes with natural glycerophospholipids and a polyphenol.

Oxidized membrane lipids are AhR ligands that have previously been implicated in the pathogenesis of SCZ (Figure 1). MLR and kaempferol exert a dual mechanism of action: the elimination of lipid peroxides and GSK-3β inhibition [179]. Replacing oxidized plasma and/or mitochondrial membrane fats with healthy natural lipids averts ferroptosis and optimizes neurotransmission by correcting membrane distortion. Conversely, oxidized membrane lipids can trigger neuronal demise by ferroptosis [180]. Indeed, MLR reverses biophysical changes in the plasma and mitochondrial membrane induced by oxidized lipids. This action is not different from that of phenothiazines, which insert themselves into the lipid bilayer, lowering the curvature of cell membranes (Figure 4). In contrast, oxidized lipids form looped structures and generate membrane curvatures and pores, which leads to cell death [181].

## 7. Phenazines and Antioxidant Phenothiazines

Phenazines are microbial metabolites produced by various soil and water microorganisms that exert antibacterial, anticancer, antimalarial, and antipsychotic properties [184].

### 7.1. Natural Phenazines 

Natural phenazines are synthesized by bacteria, including *Streptococcus species* and *Pseudomonas aeruginosa*, the latter known for generating pyocyanin (5-N-methyl-1-hydroxyphenazine), a compound with electron-shuttling properties [185,186].

Neuroprotective natural phenazines such as geranyl-phenazine, an AChE inhibitor, upregulate ACh, exerting antipsychotic effects via muscarinic receptors [187,188]. Another natural phenazine with neuroprotective functions, bara-phenazines A–G, are fused molecules with antipsychotic properties derived from *Streptomyces* sp. PU-10A [189].

### 7.2. Synthetic Phenazine Derivatives 

Synthetic phenazine derivatives consist of over 6000 compounds, exerting antimicrobial, antiparasitic, neuroprotective, anti-inflammatory, and anticancer activities. Pontemazines A and B are neuroprotective phenazines derived from *Streptomyces* sp. UT1123, which, in animal studies, rescued hippocampal neurons from glutamate cytotoxicity, highlighting their pro-cognitive properties that could benefit patients with negative SCZ symptoms [190]. Pontemazines exert antioxidant, radical-scavenging properties and inhibit lipid peroxidation, suggesting beneficial effects on SCZ [191]. Halogenated phenazines act as iron chelators and are probably helpful against ferroptosis [192,193]. We believe that pontemazines and halogenated phenazines should be assessed for their antipsychotic and anti-neurodegenerative properties (Figure 4).

From a biochemical standpoint, phenazines are almost identical to phenothiazine antipsychotics and likely possess similar properties (Figure 5). Phenothiazines are typical antipsychotic drugs utilized primarily for SCZ and SLDs that block dopaminergic transmission at the level of postsynaptic neurons. They also correct the curvature and receptor alignment on neuronal/mitochondrial surfaces, restoring signaling homeostasis (Figure 4) [182]. In contrast, oxidized lipids, toxic ceramides, and 7-ketocholesterol (7KCl) form looped structures, generating membrane curvatures and pores that may lead to neuronal death [183].

### 7.3. Antioxidant Phenothiazines and Their Derivatives 

Antioxidant phenothiazines and their derivatives have recently been developed for cancer, cardiovascular disease (CVD), *Mycobacterium leprae*, and other antibiotic-resistant microbes [194].

Phenothiazine derivatives exert anti-peroxidation properties and protect against lipid pathology and ferroptosis, suggesting their efficacy as antipsychotic drugs [195]. Phenothiazine nuclei possess hydrophobic properties that facilitate their insertion into the plasma or mitochondrial membranes [196].

Propenyl-phenothiazine is a potent antioxidant with electron-donor capabilities that likely prevents gray matter loss in patients with SCZ or SLDs. Moreover, a new category of tetracyclic and pentacyclic phenothiazines with antioxidant properties have recently been developed, suggesting their likely efficacy for cognitive impairment and negative SCZ symptoms [197,198]. Furthermore, N10-carbonyl-substituted phenothiazines were demonstrated to inhibit lipid peroxidation, suggesting their enhanced antipsychotic efficacy [100].

## 8. Mitochondrial Transfer and Transplantation

Mitochondrial transplantation experiments started in the 1980s, wherein naked organelles were co-incubated with various cell types, attempting to facilitate internalization [199]. Using HeLa cells and mesenchymal stem cells as mitochondrial sources, this transplantation technique takes only 1–2 h to supply organelles to mitochondria-depleted cells [200,201,202]. At present, mitochondrial transplantation in cardiomyocytes is possible and can be confirmed by the presence of mitochondrial DNA (mtDNA) in the heart [203,204].

Mitochondrial transplantation to rescue neurons from ferroptosis is currently possible and has been successfully performed in both animals and humans; however, to the best of our knowledge, it has not been attempted as a treatment for mental illness [205].

Rescuing the mitochondria with MLR, kaempferol, and berberine (Figure 6) is a strategy for averting GSK-3β overactivation by toxic ceramides, oxysterols, or oxidized phospholipids [130,206]. In addition, SSRIs, GJA1-20K, and CD38 signaling were demonstrated to facilitate mitochondrial transfer, emphasizing potential strategies for restoring neurometabolic homeostasis in severe mental illness and neurodegeneration [207,208].

## 9. Using AhR Antagonists as Antipsychotics

Aberrant AhR overactivation has been associated with psychosis, while several antagonists of this receptor exert antipsychotic properties. The following natural and synthetic AhR inhibitors were found to be therapeutic for SCZ:Quercetin is a natural flavonoid and plant pigment that exerts antioxidant and anticancer properties. In the CNS, quercetin is a negative allosteric modulator of GABARs as well as an enhancer of glutamatergic neurotransmission, a signaling pathway deficient in SCZ [209]. In addition, quercetin inhibits the apoptosis of cortical neurons, likely preventing gray matter loss.Apigenin is a plant-based remedy extract from *Elsholtzia rugulosa* used by traditional practitioners from Africa for treating mental illness. Aside from antagonizing the AhR, apigenin exhibits vasorelaxant, antioxidant, and antipsychotic properties [209].Alstonine is an indole alkaloid with antipsychotic effects that increases serotonergic, but not dopaminergic, signaling, possibly facilitating mitochondrial transfer [209,210].Luteolin is a natural antipsychotic that exerts its beneficial actions by reducing microglial inflammation [202]. Luteolin is currently under clinical trials for SCZ treatment (NCT05204407).

## 10. Synthetic AhR Antagonists

Synthetic AhR antagonists are anti-inflammatory and anticancer compounds that likely exert antipsychotic properties.

IK-175 (structure undisclosed) was shown to block ligand-mediated AhR activation in preclinical studies. IK-175 was recently approved for cancer treatment, and it may possess antipsychotic properties [211].

HBU651 is a novel synthetic AhR antagonist developed primarily for cancer treatment that appears to be a suitable candidate for SCZ treatment [212]. Figure 7 summarizes AhR agonists/antagonists relevant for neuropsychiatry.

## 11. Recombinant Interleukin-22

Recombinant interleukin-22 (IL-22) is a pleiotropic cytokine known for facilitating tissue regeneration and protecting the GI tract barrier. Recombinant IL-22, comprising two molecules connected by a fusion protein, exerts better efficacy with reduced systemic side effects [223].

In our previous work, we hypothesized that SCZ and SLDs may be initiated by aberrant AhR hyperactivation by endogenous or exogenous ligands, including intestinal or environmental toxins, such as LPS or plasticizers, respectively [224].

This hypothesis is supported by the following findings:SCZ is often comorbid with IBD, conditions associated with increased gut barrier permeability and microbial translocation from the GI tract into host tissues, including the brain.Translocation markers, including soluble CD14 (sCD14) and lipopolysaccharide-binding protein (LBP), are elevated in SCZ, suggesting bacterial translocation.Increased BBB permeability in SCZ enables translocated gut microbes to reach the brain.

The following are examples of pathogens triggering psychosis:The *Escherichia coli* (*E. coli*) outbreak in 2011 in Germany has been associated with cases of new-onset psychosis.New-onset psychosis, or its exacerbation, has been identified in *E. coli*-associated UTIs.

IL-22 has been successfully used to restore the integrity of the gut barrier in various conditions, including IBD, HIV, and nonalcoholic fatty liver disease [225,226]. We construe that recombinant IL-22 would be effective for SCZ by limiting the translocation of bacteria and/or their molecules.

Table 2 summarizes the major representatives from the categories discussed above with potential therapeutic properties for SCZ and SLDs.

## 12. Vehicles: Lipid Nanoparticles

The COVID-19 pandemic has accelerated the development of lipid nanoparticles (LNPs), vehicles for drug delivery. As LNPs are liposoluble, they can access specific body niches, including the brain [227].

The COVID-19 messenger RNA (mRNA) vaccines, Pfizer-BioNTech and Moderna, are incorporated in LNPs comprising four lipids: 1,2-distearoyl-sn-glycero-3-phosphocholine (DSPC), PEG, an alternative cholesterol, and ionizable lipids such as SM-102 or ALC-0315 [228]. SM-102, ALC-0315, and the alternative cholesterol are proprietary molecules and have not been revealed. However, looking at the previous LNP research, the ionizable lipids likely resemble DLin-MC3-DMA, which was approved by the Food and Drug Administration (FDA) for transthyretin-mediated amyloidosis [229,230,231]. Like phenothiazines, LNPs enter cells, including neurons, through the endocytic pathway (EP). Subsequently, LNPs travel from the early to late endosomes, but not to lysosomes because the organelle pH of 4.5–5.0 could degrade the nanoparticles. Therefore, “lysosomal escape” into the cytosol must occur from the late endosomes without interfering with autophagy, as most antipsychotic drugs do.

Utilizing LNPs not for vaccination, but as a vehicle for transporting psychotropic drugs directly to the neuronal networks could revolutionize psychopharmacology. As nano-doses of antipsychotics or mood stabilizers would be utilized for the treatment of psychotic symptoms or affective disorders, systemic adverse effects would be avoided. We surmise that LNPs would be extremely efficacious as vehicles for psychotropic drugs.

## 13. Conclusions

Antipsychotic drugs are extremely helpful for the acute symptoms of SCZ or SLDs; however, sustained recovery (measured by the ability to hold a job, go to school, raise a family, or be independent in ADLs) is rarely achieved. For this reason, the next chapter in neuropsychopharmacology will have to improve functionality rather than symptom resolution.

Over the past 70 years, it has become obvious that lowering dopaminergic transmission does not restore premorbid functions in patients with SCZ and SLDs. DA is an indispensable neurotransmitter that maintains the integrity of the brain parenchyma, a physiological function disrupted by both SCZ and its treatment.

The study of neuronal senescence-induced neuropsychiatric disorders is in its infancy but is rapidly developing, especially after the advent of age-accelerating viruses, such as HIV and COVID-19. Aberrantly activated AhR, the master regulator of cellular senescence, explains not only how gut microbes and/or their molecules trigger psychosis but also how environmental pollutants precipitate SCZ or SLDs. AhR hyperactivation likely accounts for gray matter reduction, the loss of rapid brain oscillations, and oxidized lipids in the plasma and mitochondrial membranes. The recent discovery that DA, serotonin, vitamin D3, clozapine, and melatonin are AhR ligands has opened new horizons in the management of chronic psychosis. Novel strategies, such as natural senotherapeutics, MLR, GSK-3β, ceramide inhibitors, recombinant IL-22, mitochondrial transplantation or transfer, and AhR antagonists, could improve long-term recovery in SCZ or SLDs. Senescence-associated downregulation of ACh and DA likely reduces brain volume and rapid oscillations, contributing to deficit symptoms.

The utilization of natural compounds, such as kaempferol or berberine, alone or in conjunction with LNP-delivered DA-sparing antipsychotic drugs, may improve sustained recovery in patients with severe mental illness.

## Figures and Tables

**Figure 3 ijms-25-05904-f003:**
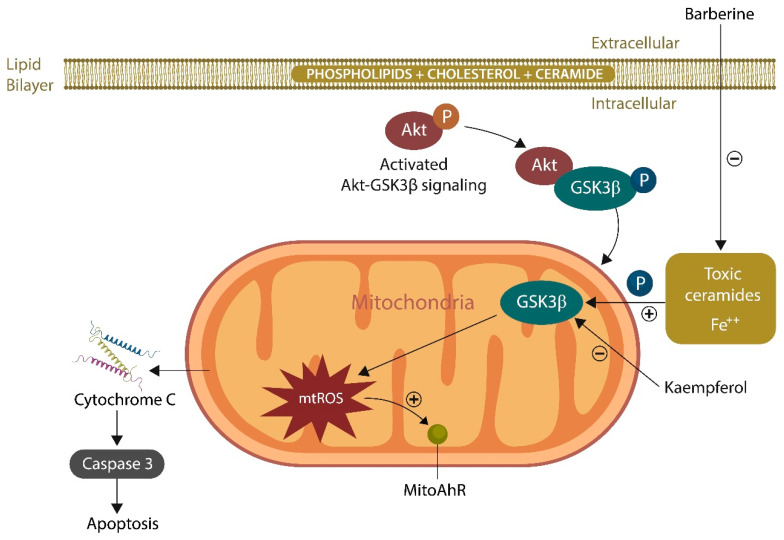
The AhR is represented in the cytosol and mitochondria (mitoAhR). Akt negatively phosphorylates GSK-3β, inhibiting its function. Toxic ceramides and iron activate GSK-3β, resulting in excessive mitochondrial ROS (mtROS) levels, which activate the mitoAhR, triggering organelle death. mtROS can also cause mitochondrial demise by activating cytochrome-C and caspase-3. The natural compounds berberine and kaempferol inhibit GSK-3β, averting organelle death.

**Figure 4 ijms-25-05904-f004:**
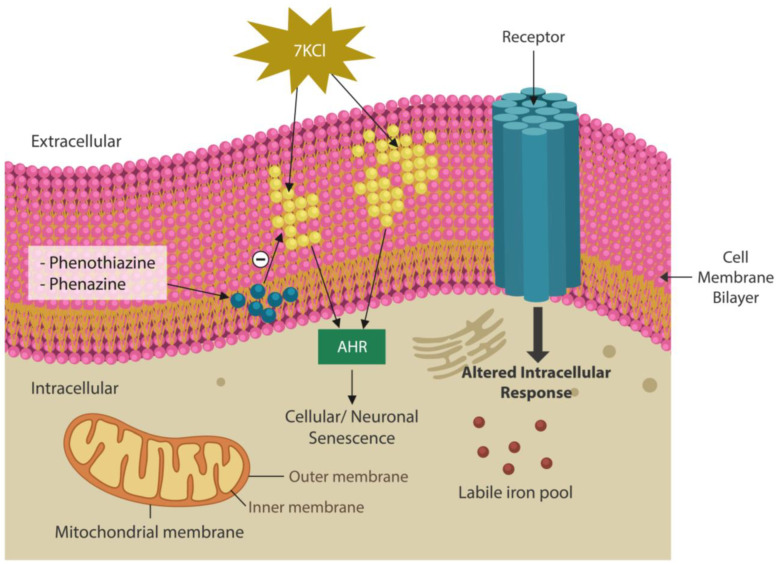
The lipid bilayer of neuronal membranes is easily oxidated when intracellular iron is upregulated [182]. Oxysterols, including 7-ketocholesterol (a toxic oxide), and oxidated phospholipids alter the biophysical properties of cell membranes, disrupting neurotransmission [183]. In addition, oxidized lipids activate the AhR, triggering premature neuronal senescence. Phenazines, phenothiazines, and their derivatives intercalate themselves into the lipid bilayer, repairing the lipids in cellular and/or mitochondrial membranes.

**Figure 5 ijms-25-05904-f005:**
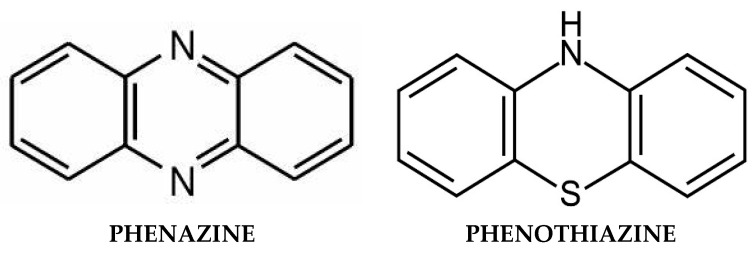
Phenazine vs. phenothiazine: similarities and differences.

**Figure 6 ijms-25-05904-f006:**
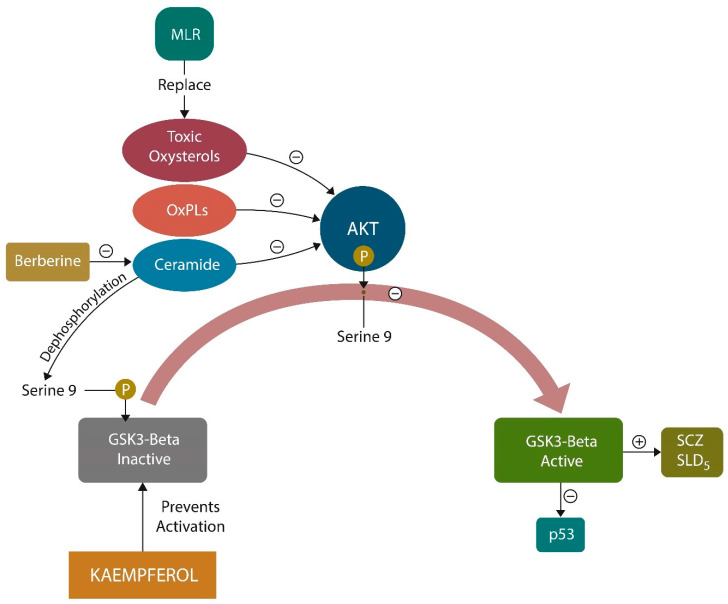
Membrane lipid replacement (MLR) replenishes oxidized lipids from the plasma and mitochondrial membrane, such as oxysterols, ceramide, and oxidized phospholipids (OxPLs) with natural glycerophospholipids. Oxidized lipids inhibit AKT (by serine-9 phosphorylation), activating GSK-3β, an enzyme associated with SCZ, SLDs, and cancer (by p53 inhibition). Berberine and kaempferol inhibit GSK-3β activation by different mechanisms, generating beneficial effects. Ceramide activates GSK-3β by the dephosphorylation of serine-9.

**Figure 7 ijms-25-05904-f007:**
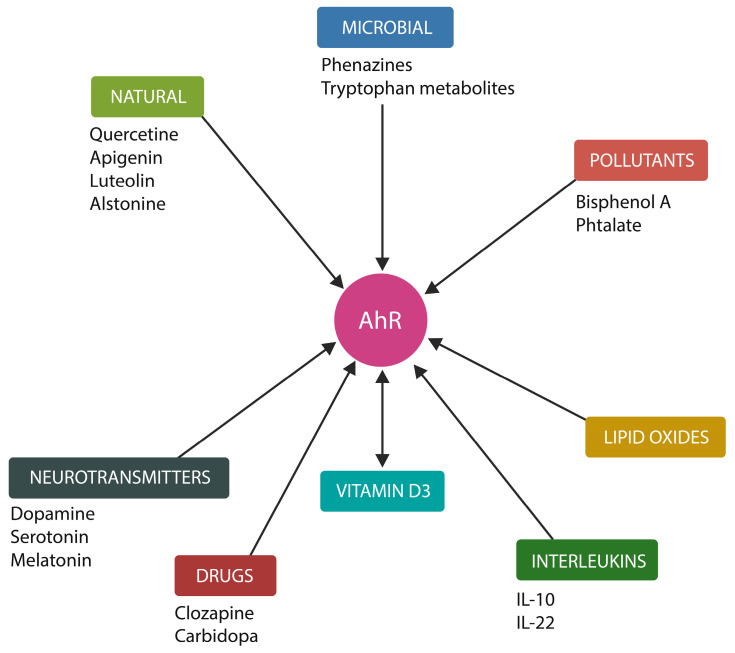
AhR agonists and antagonists relevant for neuropsychiatry [74,213,214,215,216,217,218,219,220,221,222].

**Table 1 ijms-25-05904-t001:** Natural senolytics and their sources.

Senolytics	Sources	Reference
Lycopene	Grape skin, guava, grapefruit, blueberries, and tomatoes	[166]
Apigenin	Cabbage, blueberries, and acai berries	[167]
Fisetin	Strawberries, onions, apples, mangoes, persimmons, and kiwis	[168]
Curcumin and EF24 analog	Chicken, beef, tofu, and vegetables	[169]
Epigallocatechin gallate	Apples, blackberries, broad beans, cherries, black grapes, pears, raspberries, and chocolate	[170]
Berberine	Oregon grape, phellodendron, and tree turmeric	[171]
Quercetin	Fruits, apples, onions, parsley, sage, tea, and red wine	[172]
Kaempferol	Fruits and vegetables	[173]

**Table 2 ijms-25-05904-t002:** Naturally occurring and synthetic compounds with potential benefits for SCZ and SLDs.

Compound	Naturally Occurring	Synthetic
Phenazines	Geranyl-phenazine and bara-phenazines A–G	Pontemazines A and B and halogenated phenazines
Phenothiazines		Propenyl-phenothiazine and N10-carbonyl-substituted phenothiazines
GSK-3β inhibitors	Kaempferol and curcumin	Lithium, valproic acid, clozapine, and olanzapine
AhR inhibitors	Quercetin, apigenin, alstonine, and luteolin	IK-175 and HBU651
Acid sphingomyelinase (ASM) inhibitors		Fluvoxamine, rosuvastatin, and tricyclic antidepressants
Dopamine D1R agonists		A68930, A77636, and dihydrexidine
Mitochondrial export		SSRIs
ACh agonists	*Catharanthus roseus* and *Salvia* spp. (Lamiaceae)	Cholinesterase inhibitors: donepezil, galantamine, and rivastigmine
Senotherapeutics	Please see Table 1	Senotherapeutic antibiotics
Ferroptosis inhibitors	Natural flavonoids and berberine	Fluvoxamine, SSRIs, and N acetyl cysteine (NAC)
Recombinant IL-22		
Membrane lipid replacement		
Mitochondrial transplantation		
40 HZ entrainment with sensory stimuli		

## Data Availability

Not applicable.

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
