# Peer review of "Novel Insights into Psychosis and Antipsychotic Interventions: From Managing Symptoms to Improving Outcomes"

_ijms, 2024, doi:10.3390/ijms25115904_

Round 1

Reviewer 1 Report

Comments and Suggestions for Authors

The manuscript entitled “The future of antipsychotic interventions: from managing 2 symptoms to improving outcomes” by Adonis Sfera al. for publication in this journal. I have thoroughly examined the manuscript and enthusiastically endorse it for publication in this journal.

-The research conducted is methodologically robust and of significant relevance to researchers.

-The information presented in the manuscript is original and comprehensive.

-The study aligns well with the focus of the journal.

-The conclusions drawn are well-supported by the provided data.

Minor points:

1.      Improve the title.

2.      Please check the entire manuscript for grammatical errors.

3.      Include a brief conclusion section to summarize the key findings and their implications.

4.      Go through the following recent articles and support the manuscript with the following references to make it more impressive- DOI: 10.3390/molecules27134311; 10.1021/acsami.2c22999;  10.3390/app121910130 

5.      The quality of the figures should be improved.

Comments on the Quality of English Language

 Moderate editing of English language required

Reviewer 2 Report

Comments and Suggestions for Authors

Manuscript discuss as abnormal AhR, the master regulator of cellular senescence, can explain not only how gut microbes and/or their molecules can trigger psychosis but also how environmental pollutants precipitate SCZ or SLDs.

Major

Although cell senescence has been confirmed in mental diseases such as SCZ, other studies, including genetic studies, aim to prove the neurodevelopmental hypothesis. Meanwhile, neuropathological and longitudinal studies of schizophrenia often support a neurodegenerative hypothesis. Then, it is possible to discuss the relationship between maternal gut infection (as one form stress), and AhR during pregnancy as a risk factor for schizophrenia in offspring. In fact, it is possible neurodevelopment and neurodegenerative disorder combination.

Minor

1)    Lines 136 – 138: It is not clear the relation between APOE and ferroptosis and ferrosenescence;

2)    Figures: references must be cited in the legends;

3)    Figure 3: in the legend is AhR instead Akt

4)    Figure 4: seems incomplete.. the iron and mitochondrial membranes are not shown

5)    Figure 5: who is Phenazine and who is Phenothiazine

6)    Lines 348 - 352: here is outstanding

Reviewer 3 Report

Comments and Suggestions for Authors

The authors have submitted a narrative review article of illustrating a current knowledge regarding impact of aryl hydrocarbon receptor (AhR) ligands such as dopamine, oxidized cholesterol, serotonin, clozapine, melatonin, and pollutants on premature cellular senescence which leads to schizophrenia pathogenesis in humans. To manage the disorder, it is likely to regulate signaling of AhR receptors by using the inhibitors. The authors searched a range of eligible literature, from well-known classical, and latest research regarding an association of the action of AhR ligands with pathological property of the cellular signaling function with schizophrenia and the schizophrenia-like disorders, which are primarily attributed to the discussion for a possible treatment of the disease. The authors discussed the beneficial availability of a variety of the compounds including antipsychotics which binds to the AhR receptors and the pharmacologic properties which ameliorate the states of the disease situation, resulting in possible perspectives. This issue is of interest, but impact of their narrative review is moderate. My overall concern with the review describing the current available data regarding beneficial availability of the AhR ligands listed in this review against Schizophrenia is that information provided may offer something substantial that helps advance our understanding of effective management which draws novel mechanism of action of effective antipsychotics available in clinic. The reference list may be useful for readers who are interested in this issue.

It is of importance to distinguish clearly the benefit and disadvantage of AhR ligands in terms of pathogenesis of schizophrenia in the revised version, since this is narrative so that when reading the scientific evidence presented and discussed in the text, it is very difficult to understand whether it is talking about the effectiveness of treatment or the appearance of symptoms of schizophrenia. This problem is further exacerbated by some figures such as figs 1-3 that do not clearly convey the main points of this review, so I think that the figures should be revised in line with revised main text.

Author Response

Figure 7 was introduced. It includes all the categories of compounds discussed.

The references 223 to 233 pertain to figure 7.

Clarification was made in several places (marked in yellow) for beneficial vs. detrimental effects.

Round 2

Reviewer 3 Report

Comments and Suggestions for Authors

The authors have addressed properly all the issues raised by reviewers including me. I have no more comments, and now recommend that this manuscript is acceptable for publication in the journal IJMS.